# Distribution and Expression of Pulmonary Ionocyte-Related Factors CFTR, ATP6V0D2, and ATP6V1C2 in the Lungs of Yaks at Different Ages

**DOI:** 10.3390/genes14030597

**Published:** 2023-02-27

**Authors:** Junfeng He, Yating Wei, Yan Cui, Qian Zhang

**Affiliations:** College of Veterinary Medicine, Gansu Agricultural University, Lanzhou 730070, China

**Keywords:** yak, lungs, pulmonary ionocytes, CFTR, ATP6V0D2, ATP6V1C2

## Abstract

In order to reveal the distribution and expression characteristics of the pulmonary ionocyte-related factors CFTR, ATP6V0D2, and ATP6V1C2 in the lungs of yaks of different ages. Explore the possible regulation of these pulmonary ionocyte-related factors in the yak lung for adaptation to high-altitude hypoxia. The localization and expression of CTFR, ATP6V0D2, and ATP6V1C2 in the lungs of newborn, juvenile, adult, and elderly yaks were studied using immunohistochemistry, quantitative reverse transcription PCR, and Western blotting. Immunohistochemistry showed that CFTR, ATP6V0D2 and ATP6V1C2 were mainly localized in the ciliated cells and club cells of the epithelial mucosal layer of the bronchus and its branches in the lungs. For the qRT-PCR, expression of CFTR, ATP6V0D2 and ATP6V1C2 in the yak lungs varied according to age. For Western blotting, CFTR expression in the newborn group was significantly higher than in the other three groups. ATP6V0D2 expression of the adult group was significantly higher. ATP6V1C2 expression was the highest in the juvenile group (*p* < 0.05). This study showed that ciliated cells and club cells were related to the pulmonary ionocytes in yaks. CFTR, ATP6V0D2, and ATP6V1C2 were related to adaptations of yak lungs to high altitude hypoxia, through prevention of airway damage.

## 1. Introduction

Pulmonary ionocytes are a type of cell that were first discovered in the lungs of humans and mice in 2018 [1,2]. These cells have expression profiles similar to those of ionocytes in the skin of *Xenopus laevis* and zebrafish [3,4] and account for about 1% of the composition of human and mouse tissue cell populations [1]. They are the main source of cystic-fibrosis transmembrane conductance regulator (CFTR) gene activity, as they highly express the CFTR protein [1,2] and have unique immunoreactivity to ATP6V0D2 [2]. Pulmonary ionocytes play a key role in the mechanism of pulmonary cystic fiber formation [5] and in the damage and tissue repair of many airway diseases, including asthma, chronic obstructive pulmonary disease, and bronchitis [1,2].

Cystic fibrosis (CF) is a common lethal single-gene disorder in Caucasians that is caused by a mutation of the gene on chromosome 7 encoding a transmembrane conductance regulator protein, CFTR [6]. The CFTR-encoding gene is located on the long arm of chromosome 7, is approximately 250 kb in length, and consists of 27 exons and 26 introns. The CFTR protein is composed of two globular nucleotide binding domains, nucleotide-binding domains 1 and 2 (NBD1 and NBD2) and a central regulatory region (which consists of nine phosphorylation sites of proteinase kinase A and phosphorylation sequences of other kinases such as protein kinase C/adenosine monophosphate kinase [7]) that interact with each other [8]. The abnormal function of CFTR causes many diseases, mainly manifesting as chronic sinusitis, coughing, wheezing and other types of airway inflammation, end-stage airway damage, and fibrosis of lung parenchyma, in addition to pathological phenomena such as sweating, hyperchlorhidrosis, and poor sperm release in the reproductive system [9]. Moreover, the CFTR membrane protein plays a significant role in regulating fluid transport and mucus concentration.

Vascular ATP hydrolase (V-ATPase) is a multi-subunit enzyme. As a member of the ATPase family, it is an ATP-driven proton pump, and it may be involved in regulating the secretion of pulmonary surfactants [10], in ATP hydrolysis, and in proton pump transport. The varied physiological functions of V-ATPase in different cells are established using specific subunits and subtypes [11], including the transmembrane V0 domain and the extramembrane V1 domain. V0 comprises four subunits (A, B, C, and D) for proton transfer. V1 includes eight subunits (A, B, C, D, E, F, and G) that play a role in hydrolyzing ATP to provide energy [12]. The subunit D of V0 exists as two subtypes, ATP6V0D1 and ATP6V0D2, that have 82% sequence identity [13].

The yak (*Bos grunniens*) is an endemic cattle species distributed in the Qinghai-Tibet plateau and the adjacent alpine and sub-alpine regions. In addition to China, the countries that raise yaks include Mongolia, Kyrgyzstan, Tajikistan, Bhutan, Sikkim, Afghanistan, Pakistan, and Kashmir [14]. Yaks are highly adaptable to the environment of alpine grasslands >3000 m above sea level and are extremely tolerant to severe cold, low oxygen levels, and year-round grazing conditions, forming a series of unique ecological and physiological characteristics [14,15] that are mainly manifested in the chest, heart, and well-developed lungs. Under normal conditions, the yak body obtains sufficient oxygen to adapt to the conditions of high altitude, low air pressure, and low oxygen content in the air [16]. A low-oxygen environment, like many lung diseases, can cause pulmonary fibrosis and a series of pathological changes in the body. Although the yak has been living in a harsh and low-oxygen environment for a long time, its lungs experience no pulmonary fibrosis caused by hypoxia. It is unclear whether the yak also has pulmonary ionocytes similar to those in human and mouse lungs to help adapt to the low-oxygen environment and prevent pulmonary fibrosis. Because the existing research on pulmonary ionocytes is limited to humans and mice, there are no reports for other animals living in hypoxic environments, and this lack of information includes the factors CFTR, ATP6V0D2, and ATP6V1C2. Hence, this study examined the localization and relative expression levels of CFTR, ATP6V0D2, and ATP6V1C2 in yak lungs by using immunohistochemistry, fluorescence-based quantitative PCR, and Western blotting. These methods were used to explore the possible roles of these factors in the adaptation of yak lungs to altitude hypoxia and to examine whether pulmonary ionocytes present in yak lungs are uniquely immunoreactive to ATP6V0D2. This study provides a research basis for follow-up investigations.

## 2. Materials and Methods

### 2.1. Experimental Animals and Materials

The yak lung tissues used as the research materials were collected from healthy newborn (1–7 days old), juvenile (1–2 years old), adult (3–6 years old), and elderly (7–10 years old) yaks (*n* = 3 animals per group). The lung samples of newborn yaks were from Gannan Tibetan Autonomous Prefecture, Southern Gansu Province, China. The lung samples of juvenile and elderly yaks were from Xining City, Quinghai Province, China. The lung samples of adult yaks were from Linxia Autonomous Prefecture of Gansu Province, China. After all yaks were sacrificed by carotid bloodletting, the lung tissues were quickly dissected and fixed in 4% paraformaldehyde solution for immunohistochemistry; the lung tissues for Western blotting and quantitative reverse transcription PCR (qRT-PCR) were dissected and rapidly placed in liquid nitrogen and eventually stored at −80 °C for later use.

### 2.2. Immunohistochemistry

The fixed and paraffin-embedded lung tissues were sectioned to 4 μm thickness followed by conventional deparaffinization, hydration, and antigen retrieval in citrate buffer for 15 min. The sections were then stained using the SP Kit (Beijing Biosynthesis Biotechnology, Beijing, China) according to the manufacturer’s instructions. The tissue sections were labeled with primary antibodies, including rabbit anti-CFTR primary antibodies (bs-1277, Beijing Biosynthesis Biotechnology), rabbit anti-ATP6V0D2 primary antibodies (bs-12548R, Beijing Biosynthesis Biotechnology), and rabbit anti-ATP6V1C2 primary antibodies (ab176771, Abcam) after dilution. The negative control tissue sections were incubated with 0.01 mol/L PBS instead of the primary antibodies followed by color developing using a diaminobenzidine (DAB) kit purchased from Beijing Golden Bridge Biotechnology (Beijing, China). The tissue sections were subjected to nuclear staining with hematoxylin solution followed by dehydration, clearing in xylene solution, and neutral resin mounting. After DAB color development, the sections with dark-brownish yellow staining were considered to have strong positive expression; the sections with light-brownish yellow staining were considered to have positive expression, and the sections showing the background color or colorless staining were considered as negatively stained. An Olympus BX51 microscope was used for section observation, and an Olympus DP71 Microphotography unit was used for immunohistochemical analysis.

### 2.3. qRT-PCR

The NCBI database was used to retrieve the yak’s β-actin (NM 173979.3), CFTR (NM 174018.2), ATP6V0D2 (NM 001046101.1), and ATP6V1C2 (XM 024998456.1) sequences. The Primer 6 software was used to design the primers, and these were synthesized by BGO Tech (Beijing, China; see the detailed primer sequences listed in Table 1).

The total RNA of the yak lung tissues of the four different age groups was extracted using the TRIzol reagent (Invitrogen, Carlsbad, CA, USA) according to the manufacturer’s instructions, followed by reverse transcription for cDNA synthesis. The cDNA was stored at −80 °C for later analysis. The extracted total RNA samples with optical density (OD) of 1.8–2.0 were uniformly diluted to 200 ng/μL before the reverse transcription using the corresponding Promega kits. β-actin was used as internal reference gene for the PCR of the target genes CFTR, ATP6V0D2, and ATP6V1C2 using the reaction system of 2 μL cDNA template, 0.5 μL forward primer and 0.5 μL reverse primer of the individual target gene, 10 μL Taq PCR Master Mix (C06-01003, Beijing Biosynthesis Biotechnology, Beijing, China), and 7 μL double-distilled H_2_O. The reaction conditions of the PCR were: 95 °C pre-denaturation for 3 min, 40 cycles of 95 °C denaturation for 30 s, 58 °C annealing for 30 s, 72 °C elongation for 12 s, and 4 °C for product storage. The Lightcycler96 (Roche, South San Francisco, CA, USA) was used for the fluorescence-based qRT-PCR with the reaction system of 10 μL 2×SYBR Green II PCR mix (C0-01007, Beijing Biosynthesis Biotechnology, Beijing, China), 1 μL cDNA template, 0.8 μL forward primer, 0.8 μL reverse primer of the individual target gene, and 7.4 μL double-distilled H_2_O. The reaction conditions were as follows: 95 °C pre-denaturation for 5 min, 40 cycles of 95 °C denaturation for 30 s, 58 °C annealing for 30 s, 72 °C elongation for 15 s, and 4 °C for products storage. Three replicates were set for each lung sample. According to the cycle threshold (Ct) of each sample, the 2^−ΔΔCt^ method was used to analyze the data. All data are presented as mean ± standard error.

### 2.4. Western Blotting

The frozen lung tissues of the four age groups of yaks were ground in liquid nitrogen followed by adding 0.1 g of tissue into a new centrifuge tube and subsequently adding 1 mL Radioimmunoprecipitation assay (RIPA) buffer and 10 μL PMSF. After 2.5 h shaking in an ice box, each lysate sample was centrifuged at 12,000 rpm for 10 min at 4 °C, and the supernatant was collected into a new and sterile RNase-free centrifuge tube and stored at −20 °C for later use. The extracted total protein was prepared according to the ratio of protein: 4×SDS Buffer (3:1) followed by denaturing in a metal bath for 10 min. Equal amounts of protein samples were separated by SDS-PAGE followed by transferring the protein gel to a polyvinylidene difluoride (PVDF) membrane and blocking the protein membrane with 5% skim milk. The protein membranes were independently incubated with the primary antibodies N21123 anti-β-actin mouse antibody purchased from TransGen Biotech (Beijing, China), ab2784 mouse anti-CFTR antibody (Abcam), ab236375 rabbit anti-ATP6V1C2 (Abcam), and ab176771 rabbit anti-ATP6V1C2 (Abcam) at 4 °C overnight followed by incubation with 1:5000 N21009 goat anti-mouse immunoglobulin (Ig) G (H + L) or N20915 goat anti-rabbit IgG (H + L) secondary antibodies at room temperature for 1 h before development in the chemiluminescence assay. Image J software was used to analyze the target protein bands followed by calculation of the relative protein expression based on the measured gray values. SPSS 19.0 software (IBM, Armonk, NY, USA) was used for statistical analysis in this study.

## 3. Results

### 3.1. Immunohistochemical Analysis

#### 3.1.1. CFTR Localization in Yak Lungs at Different Ages

CFTR was mainly localized in the epithelial cells, smooth muscle cells, and vascular smooth muscle cells of the pulmonary arteries of the bronchus and its branches in the lungs. It was strongly expressed in the bronchiolar epithelial cells, smooth muscle cells, and vascular smooth muscle cells of the pulmonary arteries in the newborn group (Figure 1a,b). CFTR was localized in the epithelial cells of the terminal bronchioles (Figure 1c), bronchiolar epithelial cells, smooth muscle cells, and vascular smooth muscle cells of the pulmonary arteries (Figure 1d) and was most strongly expressed in the juvenile group. CFTR was localized in the bronchiolar epithelial cells (Figure 1e) and the submucosal glands (Figure 1f) and was positively expressed in the adult group. In the elderly group, CFTR was localized in bronchiolar epithelial cells, smooth muscle cells, vascular smooth muscles of the pulmonary arteries (Figure 1g), and smooth muscle cells of terminal bronchioles and respiratory bronchioles (Figure 1h). Figure 1i–l shows the blank controls for different age groups.

#### 3.1.2. ATP6V0D2 Localization in Yak Lungs at Different Ages

ATP6V0D2 was localized and expressed in varying degrees in the yak lungs at different ages. It was mainly localized in the ciliated cells and club cells of the epithelium and mucosa of the bronchus and its branches in the lung. ATP6V0D2 was also strongly expressed in both the adult and elderly groups. In the newborn group, ATP6V0D2 was strongly expressed and localized in bronchiolar epithelial cells (Figure 2a) and epithelial cells of the terminal bronchioles (Figure 2b). In the juvenile group, ATP6V0D2 was only localized in bronchiolar epithelial cells (Figure 2c,d), showing weak-positive expression. The localization and expression of ATP6V0D2 in the elderly group were similar to those in the adult group, present in the bronchiolar epithelial cells (Figure 2e,g) and epithelial cells of the terminal bronchioles (Figure 2f,h) and having strongly positive signals in the immunohistochemistry assay. Figure 2i–l shows the blank controls of different age groups (Figure 2).

#### 3.1.3. ATP6V1C2 Localization in Yak Lungs at Different Ages

ATP6V1C2 was mainly localized in the epithelial cells of the bronchus and its branches in the lungs, the smooth muscle cells, the fibroblasts in elastic fibers, and fibroblasts in the elastic fibers of the pulmonary arteries. It was expressed in both the juvenile and adult groups. In the newborn group, ATP6V1C2 was expressed and localized in ciliated cells, club cells, fibroblasts in the elastic fibers of the mucosa of small bronchi (Figure 3a), and fibroblasts in the elastic fibers of the pulmonary arteries (Figure 3b). In the juvenile group, ATP6V1C2 was strongly expressed and localized in ciliated cells, club cells, fibroblasts in the elastic fibers of the bronchiolar mucosa (Figure 3c,e), and fibroblasts in the elastic fibers of the pulmonary arteries (Figure 3d,f). In the elderly group, ATP6V1C2 was expressed in the ciliated cells, club cells, elastic fibers in the bronchiolar mucosa, fibroblasts in the elastic fibers of the pulmonary arteries (Figure 3g), the ciliated cells and club cells in the mucosa of terminal bronchioles, and the fibroblasts in the elastic fibers (Figure 3h). Figure 3i–l shows the blank controls of different age groups.

### 3.2. Fluorescence-Based qRT-PCR

After the conventional PCR amplification, the PCR products were separated by 2% agarose gel electrophoresis to obtain a single band that matched with the expected band (Figure 4). Using qRT-PCR to detect the expression of CFTR, ATP6V0D2 and ATP6V1C2 in the yak lungs at different ages, with single melting curve peaks and primer specificity meeting the test requirements (Figure 5). The results showed that the mRNAs of CFTR, ATP6V0D2 and ATP6V1C2 were expressed to varying degrees in the yak lungs at different ages. The mRNA expression of CFTR was the highest in the adult group followed by the elderly group, and the mRNA expression of CFTR was the lowest in the yak lungs of the juvenile group. The mRNA expression of CFTR varied significantly among different ages (*p* < 0.05). The mRNA expression of ATP6V0D2 was the highest in the newborn group and showed significant differences between the newborn and the juvenile groups (*p* < 0.05) and between the juvenile and adult groups (*p* < 0.05), but not between the adult and elderly groups (*p* > 0.05). The mRNA expression of ATP6V1C2 in different age groups showed the same trend as the mRNA expression of CFTR (Figure 6).

### 3.3. Western Blotting

The expressions of CFTR, ATP6V0D2, and ATP6V1C2 proteins in the yak lungs varied at different ages (Figure 7). The CFTR protein expression of the newborn group was significantly higher than in the other three groups (*p* < 0.05). Pairwise comparisons showed no significant differences in CFTR protein expression between the juvenile, adult, and elderly groups (*p* > 0.05, Figure 7B). The ATP6V0D2 protein expression was the highest in the adult group (*p* < 0.05) followed by the newborn group, and the ATP6V0D2 protein expression was the lowest in the juvenile group (Figure 7C). The ATP6V1C2 protein expression was the highest in the newborn group (*p* < 0.05) followed by the adult group, and the ATP6V1C2 protein expression was the lowest in the juvenile group. Pairwise comparisons showed significant differences in the ATP6V1C2 protein expression among all age groups (*p* < 0.05, Figure 7D).

## 4. Discussion

### 4.1. Pulmonary Ionocytes

Pulmonary ionocytes were discovered by Danie and Linsey in the mouse and human airway tissues of the lungs. The cells play a key role in the biological mechanisms underlying the formation of pulmonary fibrosis and the repair of airway-damaged tissues, accounting for 1% of the epithelial cell population in mouse and human airways. The gene expression pattern of pulmonary ionocytes is similar to those of ionocytes in fish gills and in frogs [1,2] and is an important regulatory point for the maintenance of ion balance in fish. The main functions of pulmonary ionocytes involve Cl^−^ secretion, Na^+^ diffusion across membranes, ion absorption in low-salinity solutions, and ion secretion in high-salinity solutions [17]. Because ion transport across the epithelium is an energy-consuming process, all ionocytes require sufficient mitochondria to produce large amounts of ATP. Compared with the surrounding cells, ionocytes have a higher density of mitochondria in the cytoplasm [16]. On the basolateral membrane, ionocytes have specific transport carriers or channels to ensure the directional movement of ions. Danie believes that ionocytes express CFTR [1]. The study by Lindsey showed that these cells co-expressed multiple subunits of forkhead box L1 (Foxl-1), vacuolar-type H+-ATPa (V-ATPase), and CFTR and had unique immunoreactivity to ATP6V0D2 [2]. Studies in recent years have shown that V-ATPase and CFTR not only play important roles in the tumorigenesis and progression of many cancers, such as esophageal cancer [18], but also are involved in distal renal tubular acidosis [19].

### 4.2. CFTR

Since the discovery of CF approximately three decades ago, more than 2000 CFTR mutations and variants have been identified. CFTR transgenic animal models are currently available for mice [20], CF rabbits [21], CF pigs [22], CF sheep, and CF ferrets [21]. This study used yaks that live in a hypoxic environment as the research subjects to explore the possible relationship between CFTR and lung adaptation. Mutations in CFTR affect the normal physiological functions of the epithelial tissues of the gastrointestinal tract and the respiratory system, affecting epithelial cells of the airway, small intestine, pancreas, liver, and reproductive tracts, and are expressed in the epithelial cells at the top of the plasma membranes of the airway, digestive tract, and reproductive tract [8]. Although these diseases affect multiple organs, the most serious and life-threatening pathology occurs in the lungs. In 1992, Engelhardt et al. [23] reported that submucosal glands were the main part of the lungs expressing CFTR. Adult lungs had the highest expression levels of CFTR protein in the submucosal acini. The study of Trezise et al. [24] showed low CFTR mRNA and protein expression in the epithelial cells of human lungs. In contrast, high CFTR expression was found in the epithelial cells of the submucosal glands, and these epithelial cells were usually confined to the proximal airway supported by cartilage. These findings differed from the present results. In this study, the immunohistochemistry assays showed that CFTR was mainly localized in the ciliated and club cells in the mucosa of the bronchus and its branches and the surrounding smooth muscle cells in yak lungs. CTFR was also localized in the vascular smooth muscle cells in the pulmonary arteries. Jiang et al. [25] studied the cellular heterogeneity of pulmonary CFTR expression and function and showed that non-ciliated cell subpopulations expressed CFTR protein and mRNA at extremely high levels, suggesting a different function from ciliated cells in maintaining the composition of airway surface fluid and electrolytes. Our results showed that there was a difference between the expression levels of CTFR protein and mRNA. In terms of mRNA levels, the adult group had the highest mRNA expression of CTFR followed by the elderly group. In terms of protein levels, the newborn group had the highest protein expression of CTFR, a level that was significantly higher than in the other three groups (*p* < 0.05). No significant differences in CTFR protein expression were found between the juvenile, adult, and elderly groups in the pairwise comparisons (*p* > 0.05), suggesting that this was related to the formation of physiological structures in the yaks to adapt to the hypoxic environment. The tunica media of the pulmonary arteries of yaks at different ages are thicker than those of animals living at lower altitudes. With increasing age, the yak lungs gradually adapt to the high-altitude hypoxic environment, and the thickness of the tunica media of the pulmonary arteries gradually decreases [16]. In addition, the expression level in yaks of the newborn group was higher than in the other three groups, while only small differences in the expression level were found in the remaining three groups in both the immunohistochemistry assays and Western blotting.

### 4.3. V-ATPase

As a member of the ATPase family, V-ATPase is an ATP-driven proton pump that is widely distributed in the membranes of various types of cells (e.g., osteoclasts, goblet cells, and renal tubular epithelial cells) and organelles (e.g., lysosomes and secretory vesicles) [11]. V-ATPase plays an important role in membrane trafficking, protein degradation, viral and toxin entry, bone resorption, pH homeostasis, and tumor cell invasion [26]. Its structure includes the peripheral V1 domain that hydrolyzes ATP, the complete V0 domain that transports hydrogen ions [12,27], and the additional Ac45 protein in eukaryotes (with proton-pump activity) [26], in which the C1 subunit is commonly expressed and the C2 subunit is only detected in the kidneys and lungs. The D2 subunit is present in the kidneys, lungs, and osteoclasts [19]. V-ATPase may be involved in regulating the secretion of pulmonary surfactants and maintaining their acidic pH [9]. The different physiological functions of V-ATPase in different cell membranes are established using specific subtypes of subunits [28] that have been identified in different tissues and cell types and have different functions. In mammalian cells, the V1 and V0 domains are reversibly dissociated in response to consumption of glucose or to insect cell molting, and this may also be a means of preserving ATP in cells [26]. The results of immunohistochemistry assays in this study showed that ATP6V0D2 was commonly expressed in the epithelial cells of the bronchial branches in the yak lungs and was strongly expressed in the lungs of adult and elderly yaks. It was not specifically expressed in a single cell type. Previous studies also supported our result that ATP6V0D1 was widely expressed, while D2 was mainly expressed on the cell surface of human kidneys and osteoclasts [29]; ATP6V0D2 is also overexpressed in most cancer tissues such as in melanoma, pancreatic cancer, and kidney cancer [30]. The results of a study by Xia et al. [13] showed that ATP6V0D2 was also expressed in macrophages and was regulated by lipopolysaccharide stimulation. Nishi et al. isolated cDNA encoding subtype D from mice (D2) and found that it was mainly expressed in the kidney, lung, testis, skeletal muscles, heart, and spleen [31].

In this study, qRT-PCR showed that ATP6V0D2 had the highest mRNA expression level in the lungs of the newborn yaks, followed by juvenile yaks, adult yaks, and elderly yaks (*p* < 0.05). No significant difference in ATP6V0D2 mRNA expression was found between the adult and elderly yaks (*p* > 0.05). Western blotting showed that ATP6V0D2 was expressed to varying degrees in the yak lungs at different ages. ATP6V0D2 protein expression was the highest in the adult groups (*p* < 0.05) followed by the newborn group, the elderly group, and the juvenile group. There was a difference between the gene and protein expression of ATP6V0D2. This difference may be caused by low-level translation of genes or by rapid turnover of proteins. Similar results were found via both immunohistochemistry and Western blotting. Yang et al. [32] showed that the fetus does not breathe in the mother during the embryonic development of yaks, but rather forms a hypoxic adaptive structure, indicating that the formation of this structural characteristic is regulated by genes. ATP6V0D2 was distributed to a certain extent in the lungs of the newborn group. The juvenile period is important for structural adaptation of the heart and lungs to a high-altitude and low-oxygen environment. Our previous research showed that 3–6 months of age is critical for the development and changes of the heart and lung structures in yaks. With increasing age, the proliferation, migration, cell metabolism, and anti-damage ability of epithelial cells in the yak lungs also increased. Thus, the expression of ATP6V0D2 gradually increased and reached a maximum in the adult group in this study. As the yaks aged, the physiological functions of the lungs declined, and the expression of ATP6V0D2 was reduced. This study showed that ATP6V0D2 was localized to varying degrees in ciliated cells and club cells in the epithelial mucosal layer of the bronchioles and their branch tracheal tubes in the yak lungs. These cells in the yak lungs may have functions similar to those of pulmonary ionocytes and may be correlated with pulmonary ionocytes. However, it was not possible to detect pulmonary ionocytes in yak lungs using only ATP6V0D2. The expression of ATP6V0D2 in yaks of different ages was basically in line with the formation and mechanism of adaptive structures of yak lungs under a hypoxic environment. However, the underlying mechanism of the adaptation to a high-altitude hypoxic environment and the prevention of airway damage or repair in the yak lungs need further investigation.

Feng et al. [10] isolated rat alveolar type II cells and detected the expression of ATP6V1C2 in vitro. The in situ hybridization showed that ATP6V1C2 was expressed in bronchioles and pulmonary alveolar epithelium cells. Multiple tissue Northern blotting showed that ATP6V1C2 was uniquely expressed in the lungs, kidneys, and testis. In this study, we also confirmed that ATP6V1C2 was mainly localized in mucosal epithelial cells and smooth muscle cells of the bronchus and its branches, fibroblasts in elastic fibers, and fibroblasts in the elastic fibers of the pulmonary arteries. In addition, ATP6V1C2 was strongly expressed in the juvenile and the adult groups. A study by Sun-Wada et al. [11] reported that the V-ATPase V1 domain C2 subunit gene (ATP6V1C2) was mainly expressed in mouse lungs and kidneys. Some researchers [13] have shown that ATP6V1C2 was expressed in the kidneys and placenta in humans but not in human lungs. The currently available data also demonstrate tissue-specific expression of ATP6V1C2 in rats. ATP6V1C2 was mainly expressed in the lungs and slightly expressed in the kidneys and testis. In the yak lungs, the expression of ATP6V1C2 in different age groups had the same trend as the expression of CFTR. Both adult and elderly groups had higher expression of ATP6V1C2, especially the adult group. Western blotting results showed that ATP6V1C2 protein expression was the highest in the juvenile group (*p* < 0.05) followed by the adult group, and the ATP6V1C2 protein expression was the lowest in the newborn group. These findings may be related to the pathological characteristics of adult yak lungs without typical pulmonary fibrosis or pulmonary hypertension related to the hypoxic adaptability and physiological function of yak lungs during the entire growth process, especially in the adult group. Under regulation by various factors, the impact of hypoxia on the lung structure of yaks gradually weakened with increasing age.

## 5. Conclusions

This study is the first to investigate the pulmonary ionocyte-related factors CFTR, ATP6V0D2, and ATP6V1C2 in yak lungs. CFTR, V-ATPase subunit genes, ATP6V1C2, and ATP6V0D2 were localized in the ciliated cells and club cells of bronchioles and their branches in the lungs of yaks of different ages. The differences in the gene expression levels at different ages were consistent with the growth and development of yaks and were related to the process of adaptability to a low-oxygen environment. However, mRNA expression and the corresponding protein expression were inconsistent. The differential expression was adapted to the role of individual factors in different age groups. The above cells had the same characteristics as pulmonary ionocytes mentioned by Danie’s and Linsey’s groups but were widely distributed and were present in larger numbers. These cells may have functions similar to those of pulmonary ionocytes and thus may help prevent pulmonary fibrosis. However, the relationship between these cells and pulmonary ionocytes requires further exploration.

## Figures and Tables

**Figure 1 genes-14-00597-f001:**
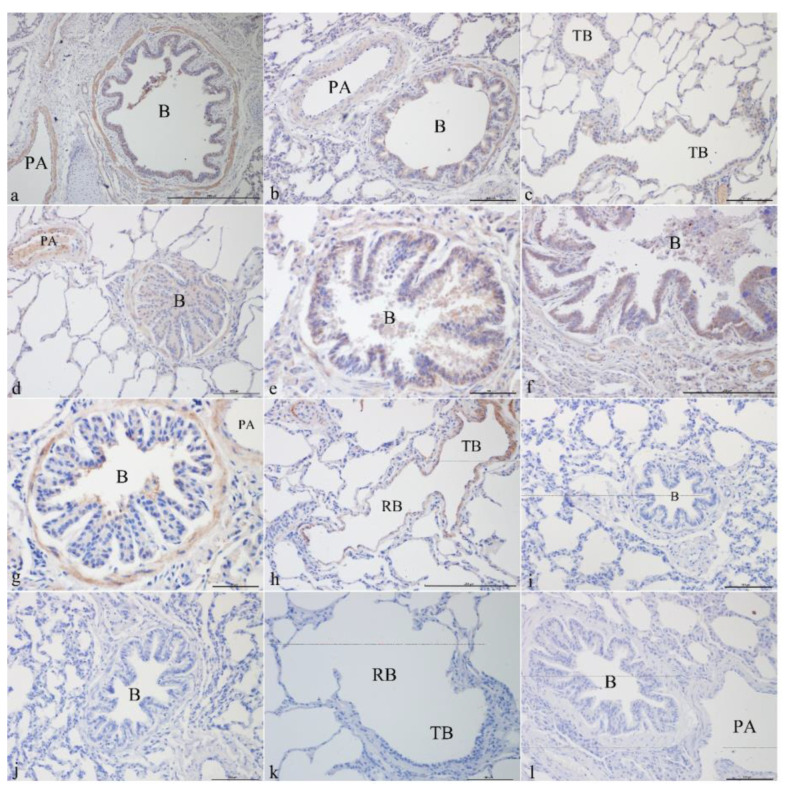
CFTR localization in yak lungs at different ages. (**a**) Bronchioles and pulmonary arteries of yak lungs in the newborn group (100×). (**b**) Bronchioles and pulmonary arteries of yak lungs in the newborn group (200×). (**c**) Terminal bronchioles of yak lungs in the juvenile group (200×). (**d**) Bronchioles and pulmonary arteries of yak lungs in the juvenile group (200×). (**e**) Bronchioles of yak lungs in the adult group (200×). (**f**) Bronchioles of yak lungs in the adult group (100×). (**g**) Bronchioles and pulmonary arteries of yak lungs in the elderly group (200×). (**h**) Terminal bronchioles and respiratory bronchioles of yak lungs in the elderly group (100×). (**i**) Blank control of bronchioles of yak lungs in the newborn group (200×). (**j**) Blank control of bronchioles of yak lungs in the juvenile group (200×). (**k**) Blank control of terminal bronchioles and respiratory bronchioles of yak lungs in the adult group (200×). (**l**) Blank control of bronchioles and pulmonary artery of yak lungs in the elderly group (200×). B—bronchiole; TB—terminal bronchiole; RB—respiratory bronchiole; PA—pulmonary artery.

**Figure 2 genes-14-00597-f002:**
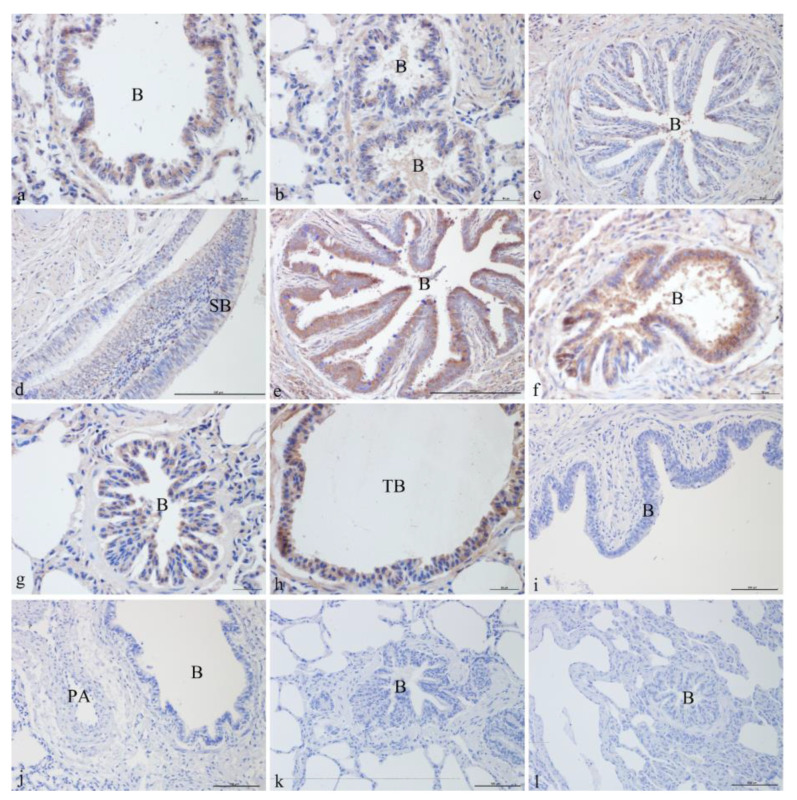
ATP6V0D2 localization in yak lungs at different ages. (**a**,**b**). Bronchioles of yak lungs in the newborn group (400×). (**c**) Bronchioles of yak lungs in the juvenile group (400×). (**d**) Small bronchi in the juvenile group (100×). (**e**) Bronchioles of yak lungs in the adult group (100×). (**f**) Bronchioles of yak lungs in the adult group (400×). (**g**) Bronchioles of yak lungs in the elderly group (400×). (**h**) Terminal bronchioles of yak lungs in the elderly group (400×). (**i**) Blank control of bronchioles of yak lungs in the newborn group (200×). (**j**) Blank control of bronchioles of yak lungs in the juvenile group (200×). (**k**) Blank control of bronchioles of yak lungs in the adult group (200×). (**l**) Blank control of bronchioles of yak lungs in the elderly group (200×). SB—small bronchi; B—bronchiole; TB—terminal bronchiole.

**Figure 3 genes-14-00597-f003:**
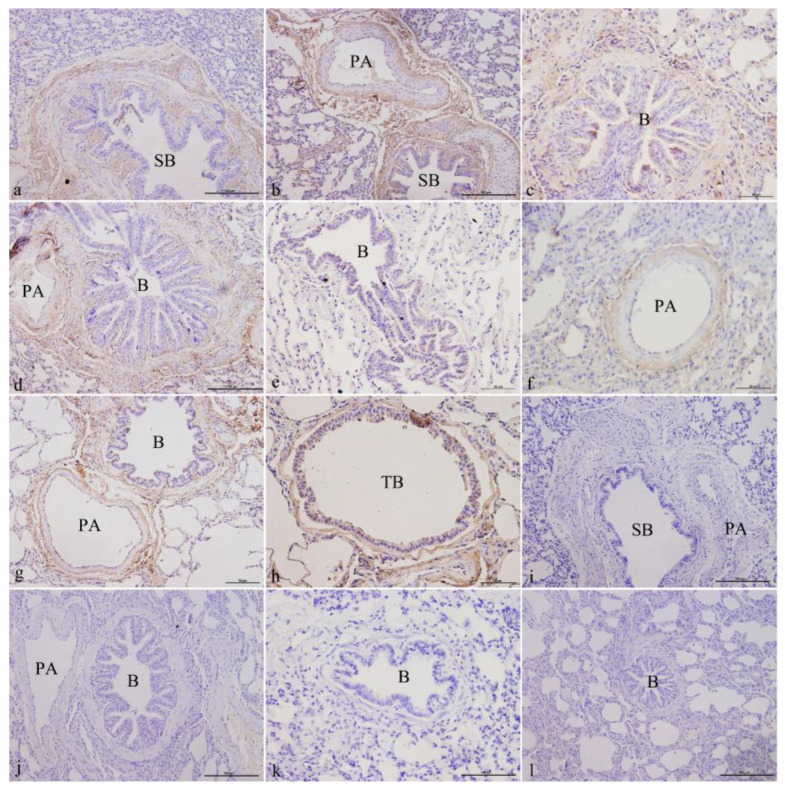
ATP6V1C2 localization in yak lungs at different ages. (**a**) Small bronchi of yak lungs in the newborn group (200×). (**b**) Small bronchi and pulmonary arteries of yak lungs in the newborn group (200×). (**c**) Bronchioles of yak lungs in the juvenile group (400×). (**d**) Bronchioles and pulmonary arteries of yak lungs in the juvenile group (400×). (**e**) Bronchioles of yak lungs in the adult group (400×). (**f**) Pulmonary arteries of yak lungs in the adult group (400×). (**g**) Bronchioles and pulmonary arteries of yak lungs in the elderly group (400×). (**h**) Terminal bronchioles of yak lungs in the elderly group (400×). (**i**) Blank control of small bronchi and pulmonary arteries of yak lungs in the newborn group (200×). (**j**) Blank control of bronchioles and pulmonary arteries of yak lungs in the juvenile group (200×). (**k**) Blank control of bronchioles of yak lungs in the adult group (200×). (**l**) Blank control of bronchioles of yak lungs in the elderly group (200×). SB—small bronchi; B—bronchiole; TB—terminal bronchiole; PA—pulmonary artery.

**Figure 4 genes-14-00597-f004:**
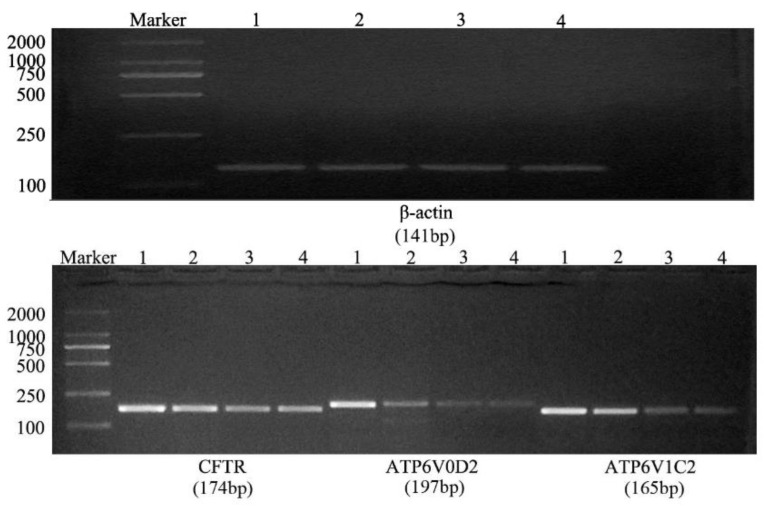
Results of conventional PCR amplification of β-actin, CFTR, ATP6V0D2, and ATP6V1C2 mRNAs. 1. Newborn group; 2. Juvenile group; 3. Adult group; and 4. Elderly group.

**Figure 5 genes-14-00597-f005:**
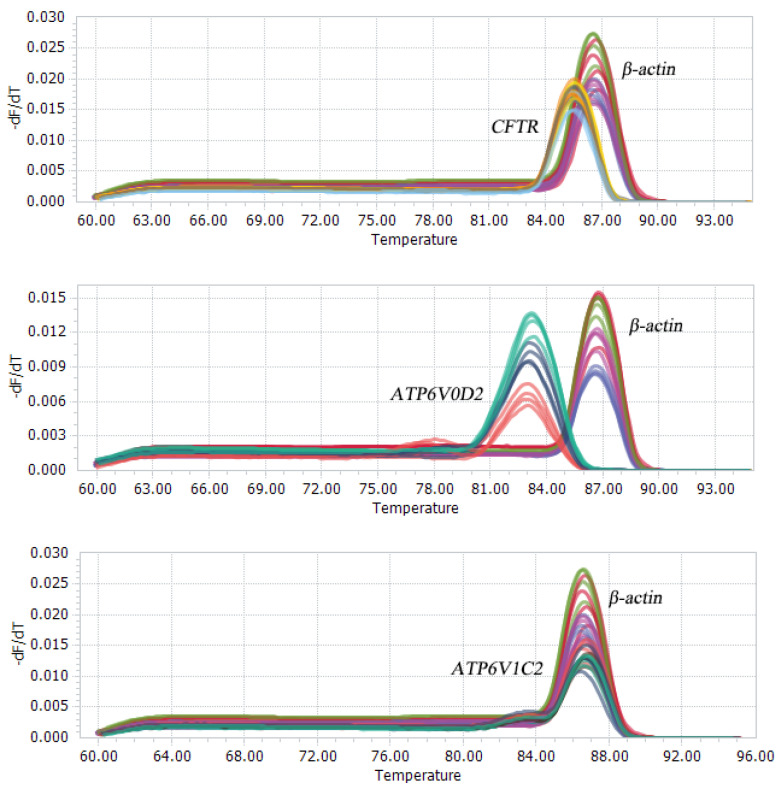
Melting curves of PCR products of CFTR, ATP6V0D2, ATP6V1C2, and β-actin mRNAs.

**Figure 6 genes-14-00597-f006:**
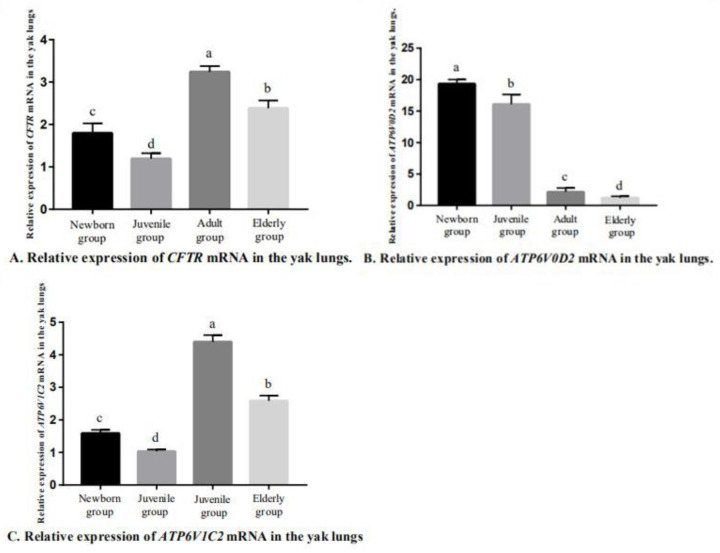
Relative expression of CFTR, ATP6V0D2, and ATP6V1C2 mRNA in yak lungs at different ages. A. Relative expression of CFTR mRNA in the yak lungs. B. Relative expression of ATP6V0D2 mRNA in the yak lungs. C. Relative expression of ATP6V1C2 mRNA in the yak lungs. Note: Different lowercase letters indicate significant differences (*p* < 0.05), and the same letters indicate non-significant differences (*p* > 0.05).

**Figure 7 genes-14-00597-f007:**
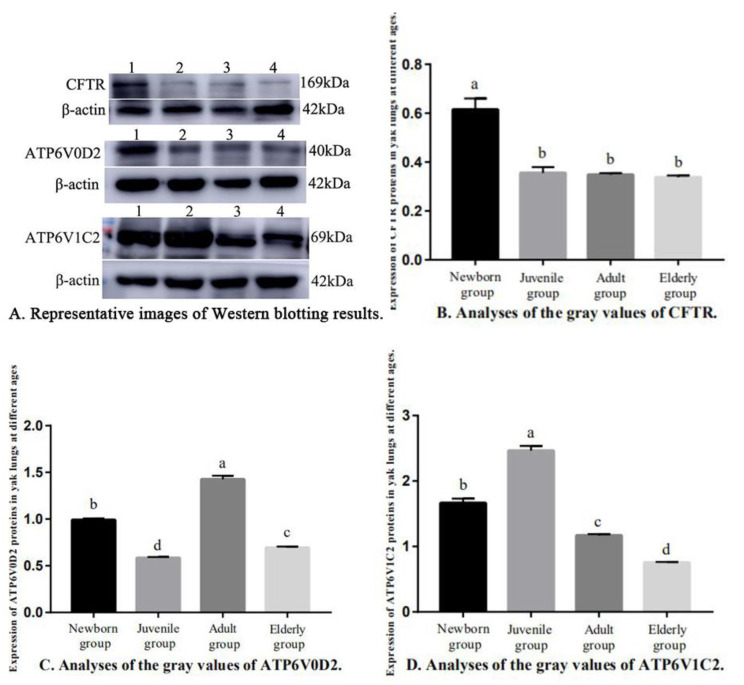
Relative Expression of CFTR, ATP6V0D2, and ATP6V1C2 proteins in yak lungs at different ages. (**A**) Representative images of Western blotting results. (**B**) Analyses of the gray values of CFTR. (**C**) Analyses of the gray values of ATP6V0D2. (**D**) Analyses of the gray values of ATP6V1C2. Note: 1. Newborn group; 2. Juvenile group; 3. Adult group; and 4. Elderly group. Different letters represent statistically significant differences (*p* < 0.05).

**Table 1 genes-14-00597-t001:** Primer information for qRT-PCR.

Primer Name	Sequence	Annealing (°C)	Length (bp)
*CFTR*	F: AAGTTGCAGATGAGGTCGGA R: GAGCACTGGGTTCATCAAGC	57.5	174
ATP6V0D2	F: ACCCCTAGCTCCGTTCTTTC R: TGATGATAAAAGCACGCCGG	57.5	197
ATP6V1C2	F: CCAAATATCCCGCCAAGCAG R: TCCGAGTGAAGAGGTTTCCC	57.5	165
β-actin	F: GCAATGAGCGGTTCC R: CCGTGTTGGCGTAGAG	60	141

## Data Availability

The authors affirm that all data necessary for confirming the conclusions of the article are present within the article, figures, and tables.

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
