# Peer review of "Distribution and Expression of Pulmonary Ionocyte-Related Factors CFTR, ATP6V0D2, and ATP6V1C2 in the Lungs of Yaks at Different Ages"

_genes, 2023, doi:10.3390/genes14030597_

Round 1

Reviewer 1 Report

The presented paper contains an interesting hypothesis worth further continuation in the form of further research. The topic is very relevant. The paper presents novel and useful findings, is sound and is well structured, and follows a logical sequence. The introduction provides evidence-based background for the research. The methodology consists of straight-forward methodologies, and deals with measurements of key parameters. The methods have been properly described. The literature is well represented, and the discussion is sufficiently elaborated for that kind of topic, with a clear (positive) conclusion. The results are well presented and data interpretation is appropriate. Results were properly reported and the findings have been accurately discussed and compared with other published papers. The findings are thoroughly discussed, and conclusions are justified by the results. The manuscript was properly conducted and findings reported are important. The paper contains valuable data. The authors investigated an interesting topic and the objective of the paper is of worldwide interest and fits well within the overall scope of the journal. I did not find any objective errors.

Author Response

Point 1: English language and style are minor spell check required.

Response 1: Checked.

Reviewer 2 Report

The article "Distribution and expression of pulmonary ionocyte-related factors CFTR, ATP6V0D2, and ATP6V1C2 in the lungs of yaks at different ages" has merit for publication in "Genes". However, I suggest to incorporate following changes in text:

1. Abstract: Lines 11-14: The sentence is incomplete and too long. Kindly complete it and split into two sentences. 

2. Line 30 & 60: Kindly italicize "Xenopus laevis" and "Bos grunniens"

3. Line 63: Kindly incorporate space "Kashmir[14]".

4. Lines 127-128: Kindly provide information on final concentration of primers and company of Taq polymerase PCR Master Mix.

5. Lines 132-133: Kindly provide information on final concentration of primers and company of 2×SYBR Green II PCR Mix. 

6. Line 136: Kindly change "production" to "products"

7. Line 143: kindly change "After 2.5-h" to "After 2.5 h"

8. Remove lines 161-163 from the text.

9. Line 164: Rename "Immunohistochemical results" as "Immunohistochemical analysis"

10. Lines 246-250: Kindly change "After the PCR amplification" to "After the conventional PCR amplification" and "with the expectation" to "with the expected band". Also split the sentence into two sentences, one for conventional PCR and other one for real-time PCR.

11. Line 261: Kindly change "Results of PCR amplification" to "Results of conventional PCR amplification" and "ATP6V1C2" to "ATP6V1C2 mRNAs"

12. Line 264: Kindly change "Melting curves of CFTR, ATP6V0D2, ATP6V1C2, and β-actin" to "Melting curves of PCR products of CFTR, ATP6V0D2, ATP6V1C2, and β-actin mRNAs"

13. Line 284: Kindly change "Rela Expression" to "Relative Expression" 

14. Lines 290, 310 and 349: Kindly correct the number of subheadings as 4.1, 4.2 and 4.3.

15. Kindly remove the lines 428-431.

16. Lines 448-449: Kindly abbreviate "Yan Cui and Qian Zhang"

Author Response

Thank you for the time you put in reviewing our paper.
